# A Workflow to Extract the Geometry and Type of Vegetated Landscape Elements from Airborne LiDAR Point Clouds

**Ine Rosier \*** , **Jan Diels, Ben Somers** and **Jos Van Orshoven**

Department of Earth and Environmental Sciences, KU Leuven, Celestijnenlaan 200E-bus 2411,
B-3001 Leuven, Belgium; jan.diels@kuleuven.be (J.D.); ben.somers@kuleuven.be (B.S.);
jos.vanorshoven@kuleuven.be (J.V.O.)
**\*** Correspondence: ine.rosier@kuleuven.be

**Abstract:** Rural European landscapes are characterized by a variety of vegetated landscape elements. Although it is often not their main function, they have the potential to affect river discharge and the frequency, extent, depth and duration of floods downstream by creating both hydrological discontinuities and connections across the landscape. Information about the extent to which individual landscape elements and their spatial location affect peak river discharge and flood frequency and severity in agricultural catchments under specific meteorological conditions is limited. This knowledge gap can partly be explained by the lack of exhaustive inventories of the presence, geometry, and hydrological traits of vegetated landscape elements (vLEs), which in turn is due to the lack of appropriate techniques and source data to produce such inventories and keep them up to date. In this paper, a multi-step methodology is proposed to delineate and classify vLEs based on LiDAR point cloud data in three study areas in Flanders, Belgium. We classified the LiDAR point cloud data into the classes 'vegetated landscape element point' and 'other' using a Random Forest model with an accuracy classification score ranging between 0.92 and 0.97. The landscape element objects were further classified into the classes 'tree object' and 'shrub object' using a Logistic Regression model with an area-based accuracy ranging between 0.34 and 0.95.

**Keywords:** landscape elements; agricultural landscapes; LiDAR point clouds; random forest; logistic regression; classification



## 1. Introduction

The European agricultural landscape is characterized by a range of landscape elements, both vegetated (vLE) like hedges and lines of trees and non-vegetated like ditches and sunken roads. vLEs include natural elements growing spontaneously across the landscape, but encompass predominantly vLEs from anthropogenic origin such as hawthorn hedges planted on plot boundaries. vLEs are often linearly shaped and are frequently situated on the border between agricultural fields where they provide a multitude of agro-ecosystem services [1]. They act as a habitat and movement corridor for animals and plant species and have a key role in the maintenance of biodiversity in intensive agricultural landscapes [2–4]. Furthermore, their typical geometrical arrangement following the edges of agricultural parcels and their association with non-vegetated elements like ditches and fences creates networks of landscape elements [5]. Such networks affect the parcel's and catchment's hydrology since they create both hydrological discontinuities and connections with variable intensity across the landscape. vLEs decrease or enhance runoff concentration and hence affect river peak discharges and the frequency, extent, depth and duration of floods downstream [6–9]. Despite this is an important topic as extensive areas throughout Europe are regularly affected by river flooding, and the prevalence of extreme rainfall events and episodes is expected to increase due to climate change [10], the magnitude and conditionalities of the effects have not been studied extensively. vLEs exert multiple effects on the water balance and transfer in an agricultural catchment. They promote



infiltration because their rooting zone is more porous than the one of the surrounding, often compacted agricultural soils [11,12] and decrease runoff by interception of rainwater, especially in the foliation period [13,14]. Further, by increasing the flow resistance, vLEs can temporarily retain more water, promote its infiltration over longer periods of time and hence reduce the surface runoff. The level of interception and flow resistance is dependent on the foliage density of vLEs [15–17].

In this paper, we make a distinction between two types of vLEs based on their foliage density near the ground surface: tree elements characterized by a leafless trunk and shrub elements having dense foliage down to the ground level (Figure 1). We define tree elements as vegetation with a crown structure and an unbranched trunk for at least 1 m above the ground (Figure 1A). These elements can appear as individual trees in the landscape, as a grove or as a tree line. Shrub elements are here defined as woody vegetation with no defined central trunk that has branches close to the ground (Figure 1B). Shrub elements include both small spherically shaped vegetation types like bushes and elongated types like hedges.

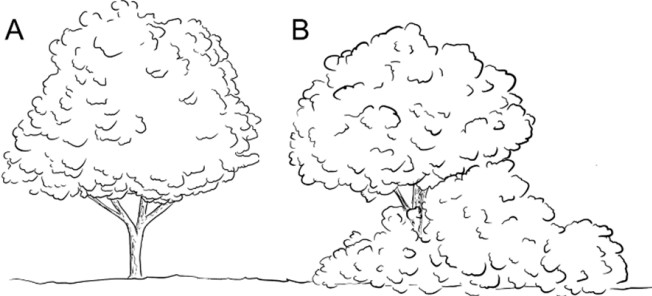

**Figure 1.** Illustration of a tree element (**A**) and a shrub element (**B**).

Few scientific reports are available about the extent to which individual vLEs and their spatial location affect peak river discharge and flood frequency and severity in agricultural catchments under specific meteorological conditions. Apart from the lack of appropriate Rainfall-Runoff models that are able to take into account the presence and characteristics of vLEs in a spatially explicit way, this knowledge gap can be explained by the lack of exhaustive inventories of the presence, geometry, and hydrological traits of vLEs, which in turn is due to the lack of appropriate techniques and source data to produce such inventories and keep them up to date. High-resolution multi-spectral imagery from airborne and spaceborne platforms which became available some two decades ago has been used to map vLEs by visual interpretation of the images backed up by time-consuming field campaigns [18,19]. Later methods were developed to automatically delineate vegetation elements from such spectral imagery [20–23]. The use of high-resolution spectral imagery for the detection of vLEs is however impeded by variations in space and time of reflectance characteristics of the elements and the occurrence of surrounding pixels with similar reflectance values [24]. Further, applications based on only spectral imagery have the disadvantage that they do not provide clear information about the three-dimensional structure of the vegetation elements which is important for hydrological studies as outlined above.

Thanks to programs producing freely available high-resolution airborne light detection and ranging (LiDAR) remote sensing data, a complement or even alternative for multi-spectral imagery is now available. LiDAR data is not only useful to map and characterize the built environment, it also showed a large potential for vegetation-related applications. The X, Y and Z coordinates and the intensity of the returned laser pulse, and the multiple return characteristics can be used to characterize vegetation vertical structure and derive various physical attributes such as leaf area [25,26]. A number of applications of airborne LiDAR data for mapping vLEs already exist. These applications however do not make a distinction between different types of vLEs [27], or generate classifications of vegetation based on their overall height and not on their vegetation density profile [28,29]. Therefore, the usability of such classifications in hydrological studies remains suboptimal.

In this paper, we describe, illustrate and evaluate an automated workflow to extract the geometric characteristics of shrub and tree-based types of vLE from airborne high-resolution LiDAR point clouds with a view to study in follow-on research not reported here, the impact of the vLE on the runoff concentration and the river discharge in agricultural catchments. It was hypothesized that the spatial and structural information derived from the LiDAR data would result in an effective identification and classification of vLEs.

## 2. Study Area and Data

### 2.1. Study Area

Three study areas were selected in Flanders, Belgium, all dominated by agricultural land use and with a total surface area of 385.92 ha (Figure 2). All three study areas show a variation in land use types, vLE density and vLE types. The first study area (SA1) is the Heulegracht catchment situated in the municipalities Gingelom and Sint-Truiden and has a surface area of 244.15 ha. In SA1, 69% is under cropland, 21% under fruit cultivation and 5% under agricultural grassland [30]. The land use of the remaining 5% consists out of forest, roads, private gardens and buildings, and flood retention installations. The area is characterized by a network of vegetative and non-vegetative landscape elements that were installed starting from 2004 to reduce soil erosion and the area's contribution to muddy floods affecting the downstream settlements [31]. Therefore, this study area is mainly characterized by rather young vLEs with a less dense vegetation cover. The second study area (SA2) is located in the municipality Huldenberg and has a surface area of 88.78 ha. In SA2, 56% of the area is under cropland and 31% is under agricultural grassland. The land use of the remaining 13% consists out of forest, roads, and private gardens and buildings. The third study area (SA3) is located in the municipality Tervuren and has a surface area of 52.99 ha. In SA3, 71% of the area is under cropland and 7% is under agricultural grassland. The land use of the remaining 22% of the area consists out of forest, roads, and private gardens and buildings. SA2 and SA3 are both part of the Langegracht catchment situated on the territory of four municipalities: Overijse, Tervuren, Bertem, and Huldenberg.

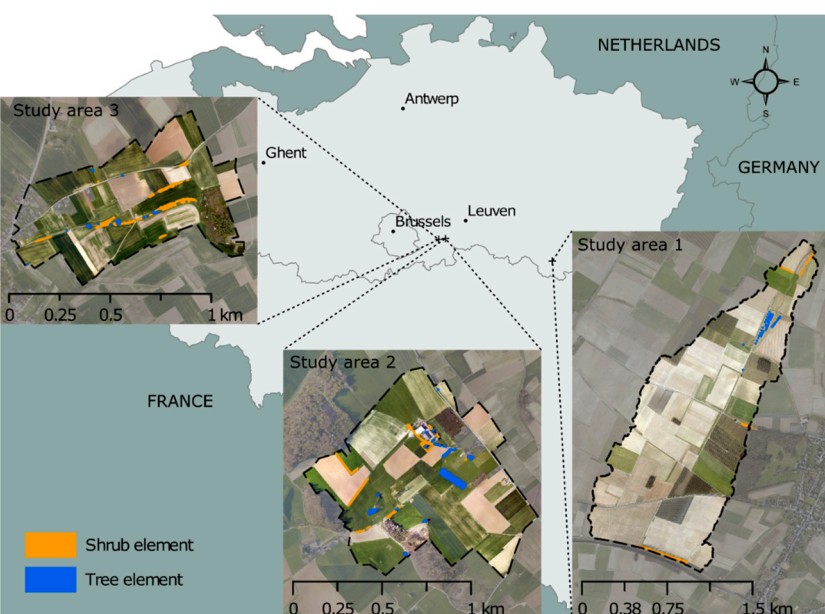

**Figure 2.** Location of the three study areas within the Flanders region, Belgium. The orange and blue polygons on the inset orthophotographs represent the shrub and tree elements measured in the field.

### 2.2. Reference Data

Reference data were collected in July and August 2019. The position and geometric characteristics of all tree and shrub elements in SA1, SA2 and SA3 were recorded to assemble a full vLE inventory. The recording was done by using a real-time kinematic (RTK)

GNSS device (Septentrio Altus APS3G) with a horizontal accuracy of 0.006 m (+0.5 ppm) and a vertical accuracy of 0.01 m (+1 ppm) [32] (Table 1). SA1 and SA3 are dominated by shrub elements while SA2 is dominated by tree elements. vLEs are more sparsely distributed over SA1 compared to SA2 and SA3. The positional data were differentially corrected by making use of the reference station network of the Flemish Positioning Service (FLEPOS) [33]. The set of recorded point locations bounding the vLEs were converted into polygons representing the vLEs (Figure 2). Finally, the geometry of the polygons was adapted by visually inspecting RGB imagery acquired simultaneously with the LiDAR data. In this way, the time lag between the acquisition of the LiDAR data and the reference data was accounted for.

**Table 1.** Aggregated surface area of the reference elements in m$^2$ and as percentage of the study area (between brackets) inventoried in SA1, SA2 and SA3.

| Study Area | Shrub Elements (m$^2$) | Tree Elements (m$^2$) |
|---|---|---|
| SA1 | 2196 (0.09) | 229 (0.01) |
| SA2 | 3774 (0.43) | 8943 (1.01) |
| SA3 | 7180 (1.36) | 601 (0.11) |

### 2.3. Airborne LiDAR and Ancillary Data

Airborne LiDAR data was collected between 4 February 2015 and 13 March 2015 in SA1, between 30 March 2014 and 5 April 2014 in SA2 and between 6 April 2014 and 12 April 2014 in SA3. The data collection was part of the DHMVII LiDAR campaign over Flanders that was carried out by Flanders Information Agency (AIV) in the winter seasons, i.e., between 15 November and 15 April of 2013, 2014 and 2015. The flight strips have a minimal forward and sideward overlap of 50% and the average point density of the final point cloud is 16 points per square meter.

The LiDAR data is freely available through the EODaS Open LiDAR portal in a LAS 1.4 format and contains for each point the X, Y, and Z coordinates, the returned intensity, the RGB values, a classification label, information on the laser pulse return number and additional scanning parameters [34]. The LAS files are provided in the standard Belgian Lambert 72 projected coordinate reference system. The Z-values are referenced to the Second General Levelling (Tweede Algemene Waterpassing, TAW), i.e., the reference for vertical height measurements in Belgium expressed in meters. The point cloud has an XY accuracy of 0.10 m and a Z accuracy of 0.05 m [35]. The laser scanner that was used can record up to five returns from each emitted laser pulse. Simultaneous acquisition of digital aerial RGB imagery with a spatial resolution of 10 cm provided RGB values for the LiDAR points.

Three ancillary datasets were used for classification of the point data: a vector dataset representing the roads within the study area, part of the High Resolution Reference Database for Flanders (Grootschalig referentiebestand, GRB) [36], a 1 m resolution Digital Elevation Model (DEM) derived from the LiDAR data [37], and a vector dataset representing the agriculture parcels and their boundaries in 2015 for SA1, and in 2014 for SA2 and SA3 [38,39].

## 3. Methodology

The workflow that was developed and applied to extract the geometric characteristics of shrub and tree-based types of vLE from airborne high-resolution LiDAR point clouds consists of seven steps (Figure 3): (1) preprocessing of the LiDAR point data, (2) characterisation of each of the LiDAR points by a set of features extracted from the neighbouring LiDAR points, (3) classification of the LiDAR points into one of two classes: vLE and other non-ground points, (4) clustering of the vLE points, (5) segmentation of the vLE points into 2D objects, (6) characterisation of the 2D objects with features extracted from the corresponding set of LiDAR points, and (7) classification of the 2D objects into the classes 'tree element' and 'shrub element'. All data were processed and analyzed using free and

open-source software. Scripting was done in Python (3.7.9), using the libraries NumPy (1.19.1), Pandas (1.1.3), and SciPy (1.5.1).

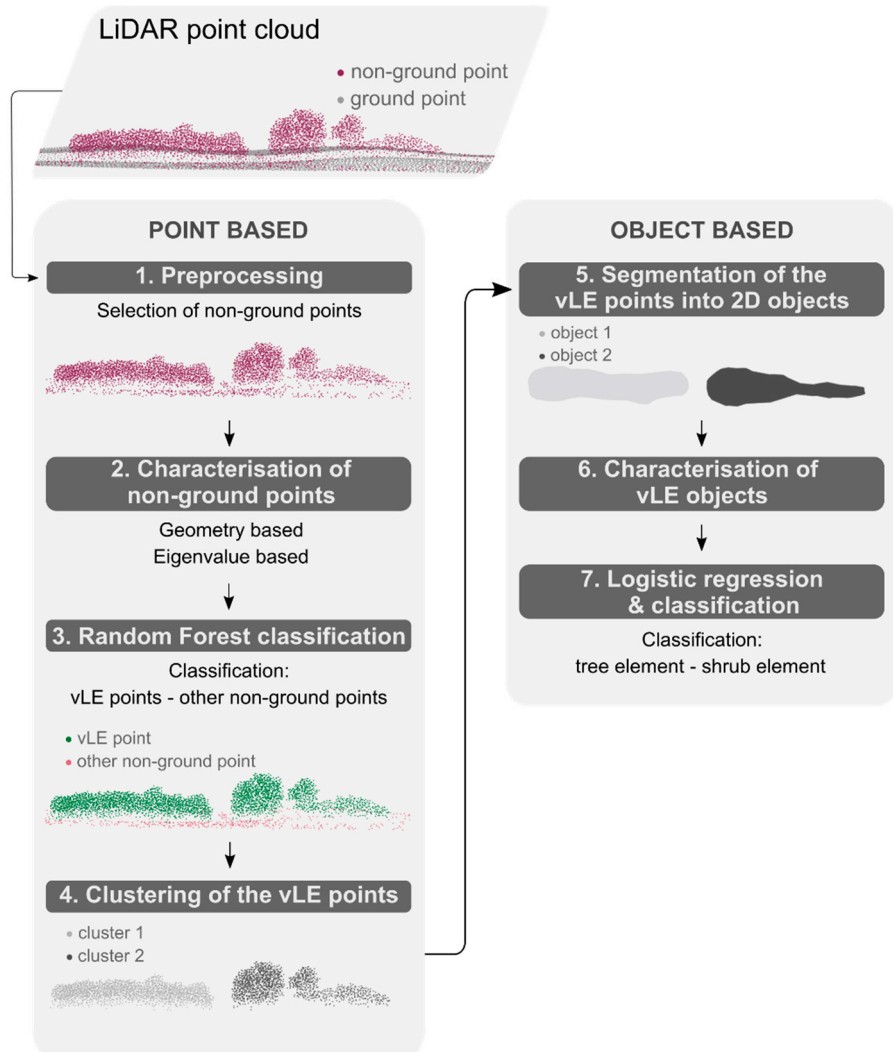

**Figure 3.** Schematic representation of the workflow applied to classify LiDAR points into vLE objects.

### 3.1. Preprocessing of the LiDAR Point Data

The LAS points received from the data provider were already classified as 'ground level', 'non-ground level' or 'water' using an iterative algorithm of the Terrascan software [35]. From this classified point cloud, all LiDAR non-ground level points were selected. Points within private gardens were omitted because of the lack of reference data. Private gardens were identified through field visits. Points belonging to wooded areas larger than 500 m$^2$ in which the trees are not planted in a linear arrangement were omitted as detection of these areas is out of the scope of this study. These wooded areas were identified from aerial RGB imagery acquired simultaneously with the LiDAR data. Lastly, LiDAR points located within fruit tree orchards were omitted because their typical multi-row planting system could cause confusion with vLEs such as tree rows. Orchards were identified from the vector dataset representing the agriculture parcels and their boundaries in 2015 for SA1, and in 2014 for SA2 and SA3 [38,39].

### 3.2. LiDAR Point Data Feature Extraction

For each point that is part of the point cloud, the number of returns, the normalized return number (i.e., sequential return number divided by the number of returns), the return intensity, the RGB values, the normalized height value (i.e., the elevation above

ground level), the shortest euclidean distance towards roads, and the euclidean distance towards agricultural parcel boundaries were considered a feature. The normalized height value (h) was determined as the height difference between each raw LiDAR point and the corresponding DEM value extracted from the 1 m resolution DHMVII product [37]. The euclidean distance towards roads and the euclidean distance towards agricultural parcel boundaries were selected because vLEs often appear alongside parcel boundaries and roads [18]. Further, seven geometry-based and eleven eigenvalue based features were calculated (Table 2). This was done by defining for each point $p_i$ its local neighborhood in the 3D space with a certain radius $r_l$, where $l \, \varepsilon \, \{1 \text{ m}, 2 \text{ m}, 5 \text{ m}, 10 \text{ m}\}$, and the k points within this neighborhood $\{n_1, n_2, \dots, n_k\}$. The radii were in line with the dimensions of the vLEs effectively present in the study areas.

Based on these neighboring points, seven geometric features were calculated: normalized height difference ($\Delta h$), standard deviation of normalized height values ($\sigma_h$), the variance of the normalized height values ($\sigma_h^2$), the mean of the normalized height values ($\bar{h}$), the minimum of the normalized height values, the maximum of the normalized height values, and local point density (D). Further, eleven eigenvalue based features were computed. The eigenvalue-based features contain information about the relative spatial arrangement of a LiDAR point and its k neighbors. To do this, the 3D covariance matrix was used to derive the eigenvalues $\lambda_j$ with $j \, \varepsilon \, \{1, 2, 3\}$, $\lambda_1 \geq \lambda_2 \geq \lambda_3 \geq 0$ and $\lambda_1 + \lambda_2 + \lambda_3 = 1$ from the 3D coordinates of the points in the k-neighborhood of $p_i$. These eigenvalues itself were used as features, and they were also used to calculate: anisotropy ($A_\lambda$), curvature ($C_\lambda$), eigentropy ($E_\lambda$), linearity ($L_\lambda$), omnivariance ($O_\lambda$), planarity ($P_\lambda$), sphericity ($S_\lambda$) and sum of eigenvalues ($\Sigma_\lambda$). This was done for each of the four radii [40].

**Table 2.** Geometry and eigenvalue based features calculated for classification of LiDAR point cloud data into vLE classes.

| Geometry Based Features | | |
|---|---|---|
| **Feature** | **Formula** | |
| Normalized height difference | $\Delta h = \max\limits_{i:1 \to k} h_i - \min\limits_{i:1 \to k} h_i$ | [41] |
| Normalized height values standard deviation | $\sigma_h = \sqrt{\frac{1}{k} \sum\limits_{i=1}^{k} \left(h_i - \bar{h}\right)^2}$ | [41] |
| Normalized height values variance | $\sigma^2_h = \frac{1}{k} \sum\limits_{i=1}^{k} \left(h_i - \bar{h}\right)^2$ | [42] |
| Normalized height values mean | $\bar{h} = \frac{1}{k} \sum\limits_{i=1}^{k} h_i$ | [43] |
| Normalized height values minimum | $\min\limits_{i:1 \to k} h_i$ | [43] |
| Normalized height values maximum | $\max\limits_{i:1 \to k} h_i$ | [43] |
| Local point density | $D = \frac{k}{\frac{4}{3}\pi r_l^3}$ | [41] |
| **EigenvalueBased Features** | | |
| **Feature** | **Formula** | |
| eigenvalue 1 | $\lambda_1$ | [42] |
| eigenvalue 2 | $\lambda_2$ | [42] |
| eigenvalue 3 | $\lambda_3$ | [42] |
| Anisotropy | $A_\lambda = \frac{\lambda_1 - \lambda_3}{\lambda_1}$ | [41] |
| Curvature | $C_\lambda = \frac{\lambda_3}{\lambda_1 + \lambda_2 + \lambda_3}$ | [41] |
| Eigentropy | $E_\lambda = -\sum\limits_{i=j}^{3} \lambda_j \ln(\lambda_j)$ | [41] |
| Linearity | $L_\lambda = \frac{\lambda_1 - \lambda_2}{\lambda_1}$ | [41] |
| Omnivariance | $O_\lambda = (\lambda_1 + \lambda_2 + \lambda_3)^{1/3}$ | [41] |
| Planarity | $P_\lambda = \frac{\lambda_2 - \lambda_3}{\lambda_1}$ | [41] |
| Sphericity | $S_\lambda = \frac{\lambda_3}{\lambda_1}$ | [41] |
| Sum of eigenvalues | $\Sigma_\lambda = \sum\limits_{j=1}^{3} \lambda_j$ | [41] |

Finally, the mean and the variance of the RGB values and the mean and the variance of the return intensity within each radius were calculated.

### 3.3. Classification of Point Data

To classify the LiDAR points into the classes 'vLE point' and 'other non-ground point', a Random Forest classifier was trained and subjected to 10-fold cross-validation. To train and validate the classifier, a ground truth dataset was created by manually labeling the LiDAR non-ground points as 'vLE point' or 'other non-ground point' by using Cloud-Compare (version 2.10.2) [44] (Table 3). This was done by comparing the reference object polygons to the point cloud data to ensure no low vegetation points underneath tree canopies were included in the vLE classes. Training and validation were done using the features of the SA1 and SA2, after which the trained Random Forest model was applied to the features of the points of SA3 in the testing phase. To check for the reproducibility of the method, this was repeated twice, using the features of either SA1 or SA2 for testing and the features of the other two study areas for training and validation.

**Table 3.** Number of non-ground LiDAR points by class.

| Study Area | vLE Points | Other Non-Ground Points |
|:---:|:---:|:---:|
| SA1 | 71161 | 264800 |
| SA2 | 336605 | 308998 |
| SA3 | 258155 | 280288 |

Because the Random Forest algorithm is constructed to minimize the overall error rate of the classification results, the prediction accuracy of the majority class affects the final model more than that of the minority class [45]. This often results in a reduced classification accuracy of undersampled classes. Therefore, sample imbalances between different classes often result in an overall decline in the performance of the Random Forest classification [46]. To account for class imbalances, we used a balanced Random Forest classifier implemented in the open-source Scikit-learn Python module [47].

The efficiency and accuracy of the Random Forest classification results were evaluated by calculating the recall (completeness), precision (correctness) and the overall accuracy of the classified points from the confusion matrix. These evaluation metrics were defined as:

$$\text{recall} = \frac{\text{TP}}{\text{TP} + \text{FN}} \tag{1}$$

$$\text{precision} = \frac{\text{TP}}{\text{TP} + \text{FP}} \tag{2}$$

$$\text{overall accuracy} = \frac{\text{TP}}{\text{TP} + \text{TN} + \text{FP} + \text{FN}} \tag{3}$$

where TP, TN, FP and FN represent respectively the number of true positives, true negatives, false positives and false negatives in the confusion matrix.

### 3.4. vLE Object Segmentation

The LiDAR points classified as vLE points by the Random Forest classifier in the testing phase were used to create spatial clusters by means of the density-based clustering algorithm DBSCAN [48]. The DBSCAN algorithm requires two input parameters, i.e., the minimum number of samples in a point cluster and the epsilon parameter. The minimum number of samples (i.e., points) in a point cluster parameter was set at 15 for SA1 and at 62 for SA2 and SA3. These values were based on the number of points per vLE in our study areas. The epsilon value represents the maximum distance between two points above which they cannot belong to the same cluster. The epsilon value was automatically estimated based on the 'knee' (point of maximum curvature) in the k-distance plot, in which k equals the minimum samples in a point cluster [49,50]. The clustered points were then converted into 2D objects (i.e., the modelled 2D objects) by applying the alpha-shape algorithm [51] per cluster of points, with the index $\alpha$ set to 0.10. This algorithm describes the shape of a finite point set, with the $\alpha$ controlling the desired level of details.

### 3.5. Extraction of Features for the vLE Objects

For all 2D vLE objects, a number of features were calculated (Table 4): the minimum, maximum and mean normalized height values (h) of the LiDAR points within the segment, the mean number of returns of the LiDAR points within the segment, the surface area of the object and the 3D point density. The 3D point density was calculated as the number of LiDAR points within the segment, divided by the area of the segment multiplied by the range of normalized height values. Further, a number of features based on the LiDAR point density distribution within a segment were calculated for each segment. To do this, a cumulative point density plot was calculated for each segment in which the cumulative point density was expressed as a function of the normalized height within the segment. The relative height at which the slope of the cumulative density plot was minimal and maximal were both used as features. Further, the normalized height at which the cumulative density equaled x, divided by the normalized height at which the cumulative density equaled y, was calculated for the combinations x = 0.5 and y = 0.1, x = 0.7 and y = 0.1, x = 0.7 and y = 0.2, x = 1.0 and y = 0.2 and x = 1.0 and y = 0.1.

**Table 4.** Candidate features for the classification of the 2D vLE objects and justification for their consideration.

| Predictor Variable | Justification | |
|---|---|---|
| Minimum normalized height ($h_{min}$) | Shrub elements likely contain LiDAR points closer to the ground surface. | |
| Maximum normalized height ($h_{max}$) | Shrub elements are likely smaller. | |
| Mean normalized height ($h_{mean}$) | Tree elements likely have a zone of low LiDAR point density near the ground surface where the trunk is located and thus a higher mean normalized elevation. | |
| Mean number of returns ($n_{mean}$) | Shrub elements likely have a higher mean number of returns because of their larger foliage density. | |
| Area | Shrub elements likely have a higher surface area. | |
| 3D point density ($P_{Dens}$) | Shrub elements have dense foliage down to the ground level while tree elements are characterized by a leafless trunk. Shrub elements thus likely have a higher 3D point density. | |
| Relative height at highest point density ($z_{Pmax}$) | The slope of the cumulative density curve is minimal (dark green) at the relative height where the point density is maximal. This relative height is more likely to be higher for tree elements. |  |
| Relative height at lowest point density ($z_{Pmin}$) | The slope of the cumulative density curve is maximal (dark red) at the relative height where the point density is minimal. This relative height is more likely to be lower for tree elements. |  |
| Elevation ratio ($z_x / z_y$) | The LiDAR point density is likely distributed differently over the elevation range of shrub elements compared to tree elements. This is indicated by the ratio of $z_x / z_y$ where $z_x$ is the elevation where the cumulative density equals x (dark red) and $z_y$ is the elevation where the cumulative density equals y (dark green). |  |

### 3.6. Classification of the vLE Objects

A Logistic Regression model was trained to calculate the probability of the 2D objects being either a shrub element or not, in which case the vLE object was classified as a tree element. A probability threshold value of 0.5 was applied to distinguish the two classes. The model was trained and validated by means of 10-fold cross-validation applied on reference 2D vLE objects of SA1 and SA2 that were created by applying the alpha-shape algorithm on the ground-truth dataset. Data normalization was performed using the sklearn StandardScaler on the training data. The StandardScaler standardizes the data by removing from each sample the mean of all training samples scaling to unit variance [52]. The correlation matrix was calculated. For pairs of features for which a correlation coefficient above 95% was calculated, one of both features was dropped. To reduce the set of candidate predictor variables, the best features were selected according to the Select Percentile feature selection model [53] with the percentile equal to 70 (SP70) and the recursive feature elimination (RFE) feature selection model [54]. The Logistic Regression model was then applied to the modelled 2D objects of SA3 to classify them as a tree element or a shrub element. The predictors were scaled by applying the sklearn StandardScaler. The scaler was trained on the reference objects. This was repeated twice, using the modelled 2D objects of either SA1 or SA2 as testing set and the reference objects of the other two study areas for training and validation.

The efficiency and accuracy of the performance of the Logistic Regression classifier were evaluated by considering the overlapping parts of the reference 2D vLE objects and the modelled 2D objects. The recall (completeness), precision (correctness) and the overall accuracy of the classified objects were calculated from the confusion matrix. These evaluation metrics were defined as:

$$\text{recall} = \frac{\text{area}_{TP}}{\text{area}_{TP} + \text{area}_{FN}} \tag{4}$$

$$\text{precision} = \frac{\text{area}_{TP}}{\text{area}_{TP} + \text{area}_{FP}} \tag{5}$$

$$\text{overall accuracy} = \frac{\text{area}_{TP}}{\text{area}_{TP} + \text{area}_{TN} + \text{area}_{FP} + \text{area}_{FN}} \tag{6}$$

in which $\text{area}_{TP}$, $\text{area}_{FP}$, $\text{area}_{FN}$ represent respectively the true positive, false positive and false negative surface areas of the classified objects in the confusion matrix.

The overall performance of the procedure was evaluated by also taking into account the non-overlapping parts of the reference 2D vLE objects and the modelled 2D objects when calculating the recall, precision, and overall accuracy. This potentially increases $\text{area}_{FP}$ and $\text{area}_{FN}$.

## 4. Results

### 4.1. Point Classification

The efficiency and accuracy of the point classification of non-ground LiDAR points as 'vLE point' or 'other non-ground point' by using a Random Forest classifier were evaluated in the testing phase by calculating the recall, precision and overall accuracy from the confusion matrix (Table 5). An overall accuracy between 0.92 and 0.97 was calculated in the testing phase over the three study areas. A recall of the vLEs between 0.86 and 0.92 and a precision of the vLE class between 0.94 and 1.00 was calculated. The classification results of the 'vLE point' class are shown in Figure 4.

**Table 5.** Confusion matrix of the Random Forest classification of non-ground points.

| Study Area | Precision | | Recall | | Overall Accuracy |
|:---:|:---:|:---:|:---:|:---:|:---:|
| | vLE | Other Non-Ground | vLE | Other Non-Ground | |
| SA1 | 0.94 | 0.98 | 0.91 | 0.98 | 0.97 |
| SA2 | 0.97 | 0.87 | 0.86 | 0.98 | 0.92 |
| SA3 | 1.00 | 0.94 | 0.92 | 1.00 | 0.96 |

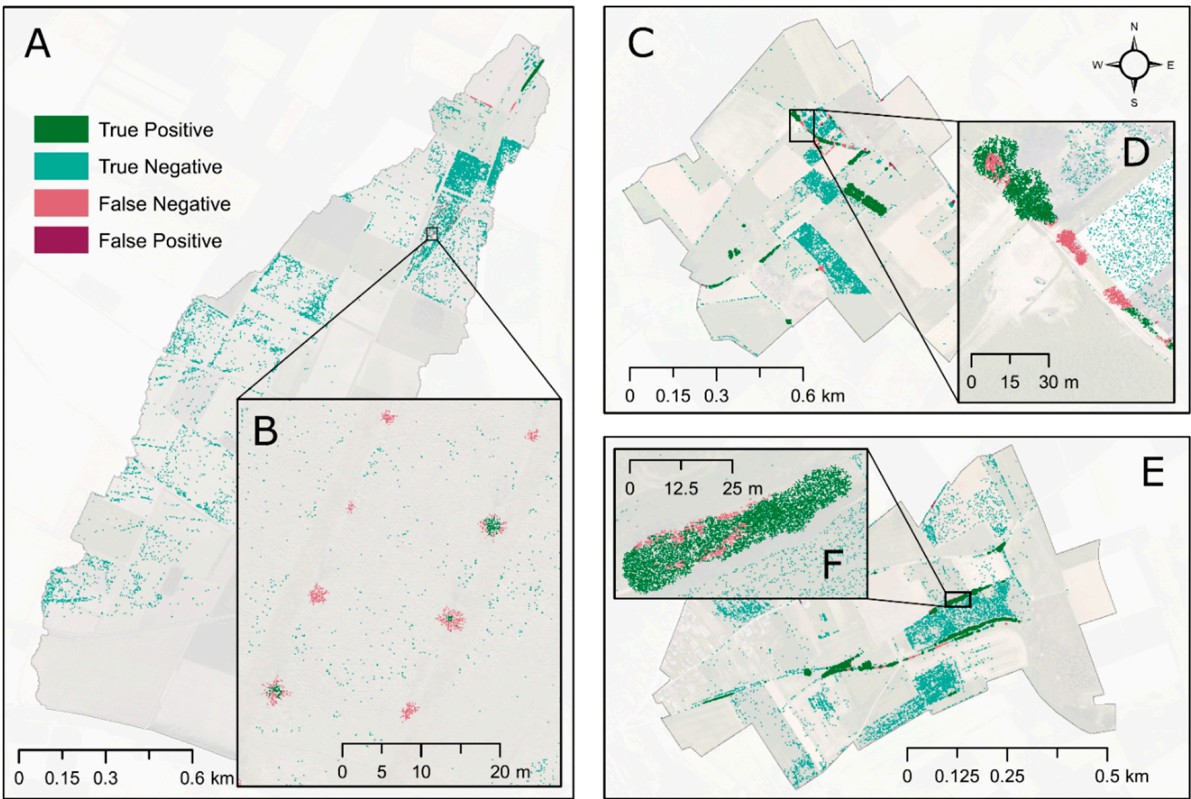

**Figure 4.** Results of the Random Forest classification of non-ground LiDAR points in SA1 (**A**,**B**), SA2 (**C**,**D**), and SA3 (**E**,**F**). The true positive class (dark green) encompasses the points correctly classified as vLE points. The true negative class (light green) encompasses the points correctly classified as other non-ground points. The false positive class (dark red) encompasses other non-ground points that were falsely classified as vLE points. The false negative class (light red) encompasses other vLE points that were falsely classified as other non-ground points.

### 4.2. Object Classification

Points that were classified as 'vLE point' were clustered by means of the density-based clustering algorithm DBSCAN and segmented into 2D objects by applying the alpha-shape algorithm [48,51]. The resulting modelled 2D objects had a total surface area of 22993.23 m$^2$, with 1605.09 m$^2$ of vLE objects modelled in SA1, 13899.37 m$^2$ modelled in SA2 and 7488.77 m$^2$ modelled in SA3. The modelled 2D objects were classified into the classes 'tree element' (value 0) and 'shrub element' (value 1) by using a Logistic Regression classifier that was trained on a set of reference objects (Table 6). The predictor variables in the Logistic Regression classifier were selected according to the SP$_{70}$ and according to the RFE feature selection approach.

**Table 6.** Predictor variables used by the Logistic Regression classifier in SA1, SA2, and SA3. X indicates a predictor variable eliminated due to correlation. Variables are defined in Table 4.

| Predictor Variable | Regression Coefficient | | | | | | | | |
|---|---|---|---|---|---|---|---|---|---|
| | **SA1** | | | **SA2** | | | **SA3** | | |
| | **/** | **RFE** | **SP$_{70}$** | **/** | **RFE** | **SP$_{70}$** | **/** | **RFE** | **SP$_{70}$** |
| $h_{min}$ | −0.12 | −0.19 | −0.17 | −1.07 | −1.07 | −1.01 | −1.20 | −1.18 | −1.11 |
| $h_{max}$ | −0.09 | | | 0.13 | | | −0.25 | | |
| $z_{Pmax}$ | 1.06 | 1.00 | 0.95 | 0.89 | 0.89 | 0.76 | 1.03 | 0.99 | 0.93 |
| $z_{Pmin}$ | −0.32 | | −0.33 | −1.32 | −1.45 | −1.60 | −0.64 | −0.64 | −0.74 |

**Table 6.** *Cont.*

| Predictor Variable | Regression Coefficient | | | | | | | | |
| | SA1 | | | SA2 | | | SA3 | | |
| | / | RFE | SP$_{70}$ | / | RFE | SP$_{70}$ | / | RFE | SP$_{70}$ |
|---|---|---|---|---|---|---|---|---|---|
| Area | 0.53 | 0.50 | | 1.04 | 0.81 | 0.77 | 0.51 | 0.48 | 0.51 |
| $h_{mean}$ | −0.68 | −0.79 | −0.92 | −0.87 | | | | −0.94 | −1.05 |
| $n_{mean}$ | 0.01 | | | 0.44 | | | −0.08 | | −0.17 |
| $P_{Dens}$ | 0.47 | 0.49 | | 0.51 | | | 0.47 | 0.52 | |
| $Z_5/z_1$ | X | X | X | X | X | X | X | X | X |
| $z_7/z_1$ | 0.78 | 0.73 | 0.75 | 0.16 | | 0.38 | 0.05 | | 0.15 |
| $z_7/z_2$ | 0.59 | 0.58 | 0.54 | 0.42 | | 0.39 | −0.02 | | −0.04 |
| $z_{10}/z_1$ | X | X | X | X | X | X | 0.23 | 0.22 | 0.14 |
| $z_{10}/z_2$ | 0.48 | 0.69 | 0.43 | 1.00 | 1.33 | 1.05 | 0.25 | 0.23 | 0.12 |

The efficiency and accuracy of the Logistic Regression based classification of the vLE objects as either 'tree element' or 'shrub element' were evaluated by considering the overlapping segments of the reference and modelled 2D objects and by calculating the recall, precision and overall accuracy from the confusion matrix (Table 7). The accuracy assessment showed a precision of between 0.13 and 0.97 for the class 'Tree element' and between 0.27 and 1.00 for the class 'Shrub element'. A recall between 0.10 and 1.00 for the class 'Tree element' and between 0.83 and 1.00 for the class 'Shrub element' was obtained. The Logistic Regression classifier had an overall accuracy of between 0.32 and 0.95.

**Table 7.** Precision, recall, and overall accuracy of the Logistic Regression model.

| Study Area | Feature Selection Model | Precision | | Recall | | Overall Accuracy |
| | | Tree Element | Shrub Element | Tree Element | Shrub Element | |
|---|---|---|---|---|---|---|
| SA1 | / | 0.13 | 1.00 | 0.98 | 0.83 | 0.83 |
| | RFE | 0.21 | 1.00 | 0.98 | 0.90 | 0.90 |
| | SP$_{70}$ | 0.14 | 1.00 | 1.00 | 0.83 | 0.83 |
| SA2 | / | 0.97 | 0.38 | 0.47 | 0.96 | 0.60 |
| | RFE | 0.89 | 0.27 | 0.10 | 0.96 | 0.32 |
| | SP$_{70}$ | 0.89 | 0.27 | 0.10 | 0.96 | 0.32 |
| SA3 | / | 0.86 | 0.95 | 0.28 | 1.00 | 0.95 |
| | RFE | 0.86 | 0.95 | 0.28 | 1.00 | 0.95 |
| | SP$_{70}$ | 0.86 | 0.95 | 0.28 | 1.00 | 0.95 |

To assess the overall performance of the workflow developed, both the overlapping and non-overlapping parts of the reference and modelled 2D vLE objects were taken into account for calculating the recall, precision, and overall accuracy (Table 8). The accuracy assessment showed a precision between 0.10 and 0.94 for the class 'Tree element' and between 0.26 and 0.77 for the class 'Shrub element'. A recall between 0.09 and 0.43 for the class 'Tree element' and between 0.57 and 0.95 for the class 'Shrub element' was calculated. Our workflow had an overall accuracy of between 0.26 and 0.73. The reference 2D objects and the modelled 2D objects classified with the Logistic Regression model using all features are shown in Figure 5.

**Table 8.** Precision, recall, and overall accuracy of the developed workflow.

| Study Area | Feature Selection Model | Precision | | Recall | | Overall Accuracy |
|---|---|---|---|---|---|---|
| | | Tree Element | Shrub Element | Tree Element | Shrub Element | |
| | / | 0.10 | 0.51 | 0.14 | 0.57 | 0.34 |
| SA1 | RFE | 0.13 | 0.53 | 0.14 | 0.61 | 0.37 |
| | SP$_{70}$ | 0.14 | 0.49 | 1.15 | 0.57 | 0.34 |
| | / | 0.94 | 0.37 | 0.43 | 0.69 | 0.49 |
| SA2 | RFE | 0.78 | 0.26 | 0.09 | 0.69 | 0.26 |
| | SP$_{70}$ | 0.78 | 0.26 | 0.09 | 0.69 | 0.26 |
| | / | 0.75 | 0.77 | 0.23 | 0.95 | 0.73 |
| SA3 | RFE | 0.75 | 0.77 | 0.23 | 0.95 | 0.73 |
| | SP$_{70}$ | 0.75 | 0.77 | 0.23 | 0.95 | 0.73 |

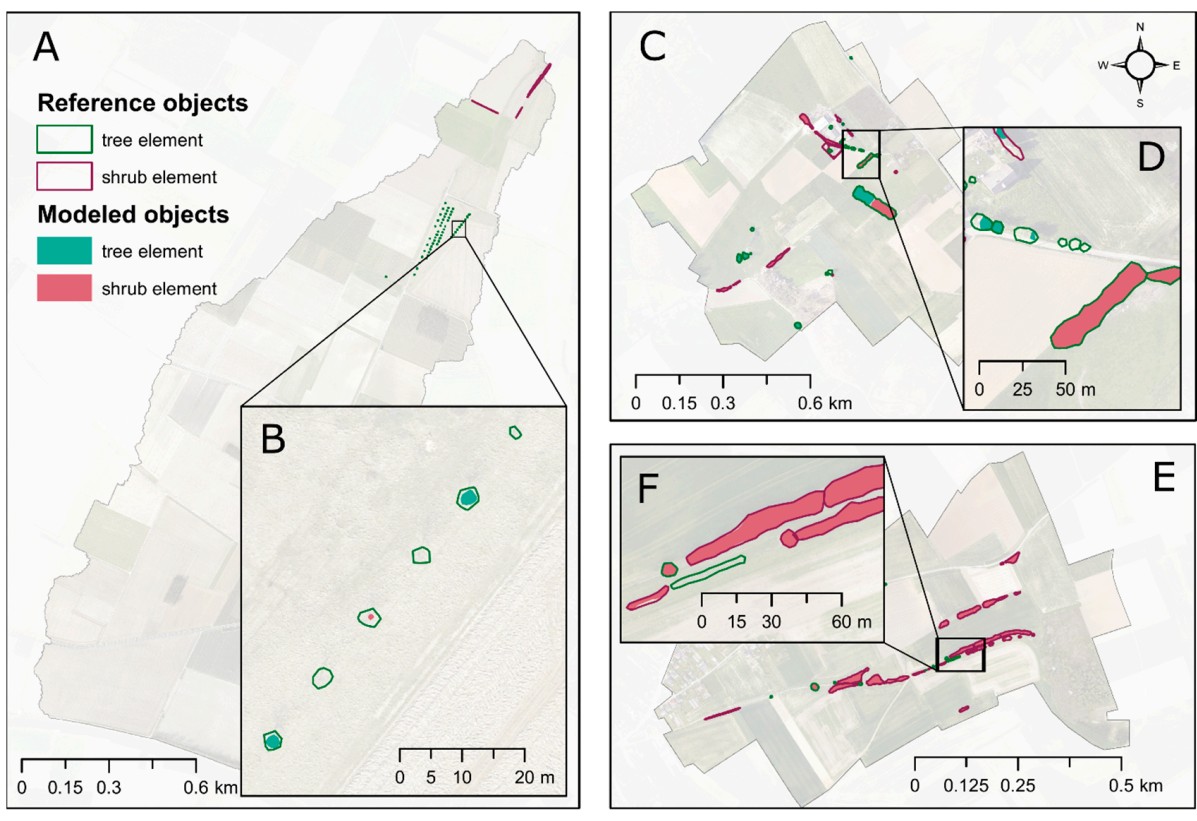

**Figure 5.** Reference 2D objects and modelled 2D objects in SA1 (**A**,**B**), SA2 (**C**,**D**), and SA3 (**E**,**F**).

## 5. Discussion

### 5.1. Classification of the LiDAR Points

A Random Forest classifier was used to classify LiDAR ground points either as 'vLE' or as 'other non-ground'. The Random Forest classifier was chosen because it has proven to be successful at classifying LiDAR points as belonging to landscape features [27,40,55]. The Random Forest classifier showed an overall accuracy between 0.92 and 0.97 in the testing phase over the three study areas. This is comparable to the accuracy of 0.97 found by Lucas et al. [27] who used a Random Forest model to distinguish vegetation LiDAR points from other LiDAR points.

A recall of the vLEs between 0.86 and 0.91 was calculated. False negative vLE LiDAR points mainly occurred at the outer ends of a vLE object. A possible explanation for this is the time lag between the recording of the LiDAR data and the ground measurements which could have resulted in reference 2D objects that are larger than they would have been

in case the reference data were recorded simultaneously with the LiDAR data acquiring. Further, the values of the neighborhood features of points at the outer ends of a vLE object are influenced by spatially nearby non-vLE ground points and can therefore be falsely classified as such since the algorithm assumes that the class labels of neighboring points are correlated [41]. Other false negative vLE LiDAR points were observed when few LiDAR data points were located within the vLE reference object. An example of this is a field with 50 young trees in SA1 (Figure 4B). Of these 50 trees, LiDAR points of only ten of these trees were classified as vLE points. The average LiDAR point count of these ten objects is noticeably higher than the average point count of the objects of which the LiDAR points were not correctly classified (212 compared to 77). This suggests that points of objects are classified more correctly when the vegetation object has a more dense point cloud and therefore classification results would improve when the LiDAR data was acquired during the leaf-on period.

A precision of the vLE class between 0.94 and 1.00 was calculated. False positive vLE LiDAR points mainly occurred at field margins. Field margins are typically characterized by a strip of weeds, especially those margins with a fence or terrace slope. Removing LiDAR points in fields with fruit tree orchards increased the precision of the vLE class from 0.10 to 0.94 in SA1 as 15.45% of this area is covered by fruit tree orchards.

### 5.2. Clustering and Segmentation of the vLE Points

The LiDAR points classified as vLE points by the Random Forest classifier in the testing phase were used to create clusters by means of the density-based clustering algorithm DBSCAN [48]. This algorithm has proven to be successful in clustering LiDAR point data in a number of applications [56–58]. These clusters were subsequently converted into 2D objects (i.e., the modelled objects) by applying the alpha-shape algorithm [51] per cluster of points, with the index $\alpha$ set to 0.10. This resulted in 18, 34, and 12 2D modelled objects for SA1, SA2, and SA3 respectively, while 62, 36, and 40 reference objects were identified in SA1, SA2, and SA3 respectively. In SA1, the difference between the number of modelled and reference objects can be explained by the Random Forest algorithm incorrectly classifying vLE LiDAR points as other non-ground points. When overlaying the modelled and reference objects, 46.43% of the total area of the reference objects of SA1 is missing in the modelled objects. For SA2 and SA3, the surface area of the reference objects missing in the modelled object dataset is only 14.95% and 6.16% respectively. The difference between the number of reference and modelled objects in SA3 is primarily the result of several reference objects being merged into one larger modelled object.

In SA1, the general flow direction goes from southwest to northeast and the modelled 2D objects are mostly positioned along this flow direction. In SA2, the general flow direction goes from northeast to southwest. The modelled 2D objects are position both alongside the flow direction and perpendicular to it. In SA3, the general flow direction goes from west to east. The modelled 2D objects are position both alongside the flow direction and perpendicular to it, creating a natural border between higher positioned agricultural parcels and the stream flow path. The position of the vLE objects along these flow paths indicates that they can be of importance in the catchment's hydrology and that it is meaningful to make a distinction between different types of vLEs that can have a distinct effect on the hydrological cycle. Previously developed applications do not make this distinction between different types of vLEs [27,59].

### 5.3. Classification of the 2D Objects

A set of features was calculated for each 2D vLE object (Table 4). These features are based on the geometry of the 2D object, the geometric and radiometric characteristics of the LiDAR points within the 2D objects, and the 3D point distribution of the LiDAR points within the 2D object. A Logistic Regression model was trained to calculate the probability of the 2D objects being either a shrub element or not, in which case the vLE object was classified as a tree element. A probability threshold value of 0.5 was applied

to distinguish the two classes. This Logistic Regression model was applied by using all features or a set of features selected by using the $SP_{70}$ and RFE feature selection models. The selected features varied between the two feature selection models and the three study areas (Table 6). The features $Z_{Pmax}$ and $Z_{10}/Z_2$ were selected in all three study areas using both feature selection methods while the overall point density ($P_{Dens}$) was only selected using the RFE feature selection model in SA1 and SA3. This shows the importance of the point distribution in the 3D space when distinguishing between tree and shrub objects. The same is observed when trying to distinguish different tree species [60]. The feature $h_{min}$ was also selected in all three study areas using both feature selection models while $h_{max}$ was omitted in all cases. This indicates there is no clear difference between the height of shrub and tree objects while the height at which foliage starts to grow differs between both object types.

To assess the performance of the classification, the classes of the overlapping segments of the reference and modelled 2D objects were analyzed (Table 7). In SA1, the Logistic Regression classification performed best when only the features selected by the RFE feature selection model were used. However, for all three models, the precision for the 'Tree element' class is low (0.21) meaning a high rate of false positive tree elements was modelled. Because of the large portion of shrub elements in SA1 (90.56% of the total surface area of all reference 2D objects in SA1), this did not translate to a low recall of the 'Shrub element' class or a low overall accuracy. A Logistic Regression model using all features or the features selected using the $SP_{70}$ feature selection model resulted in a slightly lower overall accuracy (0.83 compared to 0.90). In SA2, the Logistic Regression model performed best when all features were used with an overall accuracy of 0.60. The overall accuracy decreased to 0.32 when features were selected by using the $SP_{70}$ or RFE feature selection models. In SA2, a low precision for the 'Shrub element' class and a low recall for the 'Tree element' class could be observed when all features were used in the Logistic Regression model (0.38 and 0.47 respectively), meaning a high rate of false positive shrub elements and a high rate of false negative tree elements were modelled. The majority of the reference objects used for training of the tree element class in SA2 (50 out of 64 tree element objects) were young trees that were planted shortly before acquiring the LiDAR data as part of an erosion mitigation project in SA1. As these trees have different characteristics compared to the tree elements located in SA2, this likely influences the classification accuracy of the tree objects in SA2. This finding confirms the importance of a reference dataset that is representative of all objects that are being identified and classified [61]. In SA3, no difference in the performance and accuracy of the Logistic Regression model was found when all or only a selection of the features were used. The overall accuracy of the Logistic Regression models was calculated to be 0.95. The precision for the 'Tree element' class was low (0.28) meaning a high rate of false positive tree elements was modelled in SA3. Because of the large portion of shrub elements in SA3 (92.27% of the total surface area of all reference 2D objects), this did not translate to a low recall of the 'Shrub element' class or a low overall accuracy, as was the case for SA1.

For each Logistic Regression model, the regression coefficients of the predictor variables were calculated (Table 6). The absolute value of a regression coefficient gives an indication of the relative importance of the predictor variable this regression coefficient is linked to. A higher value for predictor variables linked with a negative Logistic Regression coefficient indicates a lower probability of the object being a shrub element, and therefore a higher probability of the object being a tree element. A higher value for a predictor variable linked with a positive Logistic Regression coefficient indicates a higher probability of the object being a shrub element, and therefore a lower probability of the object being a tree element. In general, the sign of the regression coefficients linked with the predictor variables is identical in the three study areas, especially for predictor variables with a high absolute value of the linked regression coefficient. The absolute values of the regression coefficients and therefore the importance of the predictor variables differ between the three study areas, however. In SA1, the predictor variables '$Z_{Pmax}$', '$h_{mean}$'

and '$Z_7/Z_1$' have the highest absolute values. In SA2, the predictor variables '$Z_{Pmin}$', '$h_{min}$', and '$Z_{10}/Z_2$' have the highest absolute values. In SA3, the predictor variables '$Z_{Pmax}$', '$h_{min}$', and '$h_{mean}$' have the highest absolute values. The differences in importance of the predictor variables between the three study areas can be explained by the differences in the characteristics of the vLE objects in the study areas. In SA1 for example, the vLE objects are less established and are therefore smaller and less dense. The predictor variables '$h_{min}$', '$h_{mean}$', and '$Z_{Pmin}$' are linked with a negative regression coefficient for all Logistic Regression models in the three study areas. The negative correlation coefficients for the predictor variables '$h_{min}$' and '$h_{mean}$' are in line with the expected correlation in Table 4 as shrub elements are expected to have lower minimum and mean normalized height values. The negative correlation coefficient for the predictor variable '$Z_{Pmin}$' is however not in line with the expected correlation in Table 4. It was hypothesized that the relative height at the lowest point density would be lower for tree elements as they are characterized by a leafless trunk and have therefore a low point density near the ground surface. The negative correlation between the predictor variable '$Z_{Pmin}$' and the probability of the object being a shrub element can be explained by the high biomass density of shrub elements. LiDAR pulses are more attenuated at penetrating vLE objects with a high biomass density and therefore the laser beam might not reach the lower portion of the object. This has previously been described in the context of potential errors in LiDAR-derived DEMs [62,63]. This explanation is confirmed by the LiDAR ground point density underneath the reference objects calculated as the average ground point density, which is 13.07 points/m$^2$ for shrub elements and 20.78 points/m$^2$ for tree elements suggesting the laser pulse is less likely to penetrate the object down to the ground surface for shrub elements.

The predictor variables '$Z_{Pmax}$', 'area', and '$P_{Dens}$' are linked with a positive regression coefficient for all Logistic Regression models in the three study areas. The positive correlation coefficients for the predictor variables 'area' and '$P_{Dens}$' are in line with the expected correlation in Table 4 as shrub elements are expected to have a larger area and 3D point density. The positive correlation coefficient for the predictor variable '$Z_{Pmax}$' is however not in line with the expected correlation in Table 4. It was hypothesized that the relative height at the highest point density would be lower for shrub elements since tree elements are characterized by a leafless trunk and have therefore a low point density near the ground surface. The positive correlation coefficient for '$Z_{Pmax}$' could again be explained by the dense biomass of shrub elements that is difficult to penetrate with the laser pulse [62,63]. The sign of the regression coefficient for the predictor variables '$h_{max}$' and '$n_{mean}$' was inconsistent for the three study areas. A negative correlation coefficient for '$h_{max}$' was calculated in SA1 and SA3, which is in line with the expected correlation in Table 4, while a positive correlation coefficient was calculated in SA2. A positive correlation coefficient for '$n_{mean}$' was calculated in SA1 and SA2, which is in line with the expected correlation in Table 4, while a negative correlation coefficient was calculated in SA3.

For the delineation and classification of vLE objects in areas with variations in vLE characteristics, the training dataset should be large enough to be representative of the whole area. Recalibration of the logistic regression model is needed when the model is applied to a new area. This study aids in proving the relevance of making a distinction between different types of vLE objects with unique structural characteristics.

## 6. Conclusions

We developed a seven step workflow to delineate and characterize vLEs in agricultural landscapes from airborne LiDAR data and applied and evaluated it to three study areas in Flanders, Belgium. Our method performed well when classifying LiDAR points into the classes 'vLE object' and 'other non-ground point' with an overall accuracy between 0.92 and 0.97. The importance of an extensive reference dataset was shown when classifying the modelled 2D objects. Classifying the 2D objects by means of a Logistic Regression model showed an accuracy between 0.32 and 0.95. The overall performance assessment of our workflow showed an accuracy between 0.26 and 0.73. Our workflow is based on

high-resolution LIDAR data which has the advantage that it can be used to characterize the vegetation structure. This information is essential in many applications, including hydrological modelling. Further, we showed that with a limited amount of reference objects, our workflow can be applied in an additional study area. However, classification of the 2D objects should be done in areas with high intra-class homogeneity or the Logistic Regression model should be calibrated per study area in order to classify various vLE types. Our workflow shows that there are differences in the structural composition of the considered vLE types. This proves the importance of considering different vLE types, especially in a hydrological context, as they are often located parallel and perpendicular to flow paths. The developed workflow can be used for supplementing field inventories of vLEs in agricultural landscapes and can aid in keeping existing datasets of landscape elements up-to-date. Additionally, information on the type of vLE can be used in other fields of study like hydrological and ecological modelling. Further, the developed workflow can be adapted to identify other types of vLEs in agricultural landscapes.

**Author Contributions:** Conceptualization, I.R., J.D., B.S. and J.V.O.; methodology, I.R.; software, I.R.; validation, I.R.; formal analysis, I.R.; writing—original draft preparation, I.R.; writing—review and editing, I.R., J.D., B.S. and J.V.O. All authors have read and agreed to the published version of the manuscript.

**Funding:** This research was funded by Fonds Wetenschappelijk Onderzoek (FWO), grant number 1SB6819N.

**Institutional Review Board Statement:** Not applicable.

**Informed Consent Statement:** Not applicable.

**Data Availability Statement:** The reference data presented in this study is openly available through https://github.com/IneRosier/vLE_Extraction_from_LiDARPointCloudData accessed on 27 August 2021.

**Acknowledgments:** The authors would like to thank Ann Cuppens and Rob Hillen for helping with the data collection for this study.

**Conflicts of Interest:** The authors declare no conflict of interest.

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
