# Peer review of "A Workflow to Extract the Geometry and Type of Vegetated Landscape Elements from Airborne LiDAR Point Clouds"

_remotesensing, doi:10.3390/rs13204031_

Round 1
Reviewer 1 Report
This paper studies the application of airborne LiDAR point clouds. The topic involves the current hot-spot in remote sensing research, and it exhibits that the team comprehensively understand the academic trends in this field. The three study areas selected in this paper have their own characteristics and are representative to some extent and the preprocessing of validation data set is considered comprehensively. An automatic program for extracting vegetation information based on LIDAR point cloud data is proposed as a seven step workflow, which is described in detail with reliable theory. All in all, the paper has its clear idea and reasonable structure. It provides a valuable reference for the research in this field in the specific study area and has important theoretical significance.
However, there’s still room for improvement and some suggestions are listed as follows for reference.
L20. What’s the specific accuracy index used? Please point it out at the beginning.
L54. Please explain the Figure 1A&B separately, e.g. shrub elements (Figure 1B).
L98. Change the chapter title to let the “study area” included.
L178. Please adjust the location of Figure 3. maybe forward to make it closer to its interpretation.
L180. Why describe the workflow as a part of Chapter 2, such as 2.5 to 2.10, rather than a separate chapter? Revise the structure to help the reader understand the main point more clearly.
L224. Please adjust the layout of Table 2 to make it coherent.
L318. Please make the lower right corner marks clear.
L491. Why not provide line503-506 in table format inside this paragraph?
L549. Please add some conclusion of the outlook for this study to make the paper more complete.
Author Response
We would like to thank the reviewer for their detailed comments and suggestions. We have addressed them and adapted our manuscript accordingly. Below can be found a point-by-point response to the reviewers’ comments and suggestions.
Comment 1: L20. What’s the specific accuracy index used? Please point it out at the beginning.
Reply: We adjusted the text to include the specific accuracy indices that were used, i.e. the accuracy classification score for the random forest model (line 20), and the area-based accuracy for the logistic regression model (line 22).
Comment 2: L54. Please explain the Figure 1A&B separately, e.g. shrub elements (Figure 1B).
Reply: The text was adjusted to explain Figure 1A and Figure 1B separately (line 54-60).
Comment 3: L98. Change the chapter title to let the “study area” included.
Reply: The section title was changed to “2. Study area and data” (line 100).
Comment 4: L178. Please adjust the location of Figure 3. maybe forward to make it closer to its interpretation.
Reply: The size of Figure 1 and Figure 2 was slightly adapted so that Figure 3 is positioned closer to its interpretation.
Comment 5: L180. Why describe the workflow as a part of Chapter 2, such as 2.5 to 2.10, rather than a separate chapter? Revise the structure to help the reader understand the main point more clearly.
Reply: An extra section was added: “3. Methodology” to improve the structure of the text (line 172).
Comment 6: L224. Please adjust the layout of Table 2 to make it coherent.
Reply: Table 2 was adjusted to make it more coherent.
Comment 7: L318. Please make the lower right corner marks clear.
Reply: The font size of the subscripts was changed to improve the readability (line 318 and line 322-323).
Comment 8: L491. Why not provide line503-506 in table format inside this paragraph?
Reply: Thank you for this suggestion. The text in these lines was however already shown in Table 6 of section 4.2.
Comment 9: L549. Please add some conclusion of the outlook for this study to make the paper more complete.
Reply: The conclusion section was adapted to also include the outlook for further research along the lines of this paper (line 572-576).
Reviewer 2 Report
First of all, I offer my thanks to authors for providing an excellent manuscript particularly during Covid-19 pandemic. The research topic is interesting and important . The manuscript has been organized clearly and logically. Figures and tables are clearly presented. The manuscript is worth publishing.
Authors have clearly demonstrated motivation, research questions and gap of literature with respect to the applications. However, I have some concerns regarding workflow of this paper(Figure 3, page 6). I was wondering if the authors could highlight:
(1) Problem statement
(2) Research hypothesis
(3) Limitation and advantages
Many thanks for your valuable research,
Good luck
Author Response
We would like to thank the reviewer for their time to assess our manuscript. We have addressed the suggestions made by the reviewer. Below can be found a response to the reviewers’ comments and suggestions.
Comment 1: I was wondering if the authors could highlight: (1) Problem statement; (2) Research hypothesis; (3) Limitation and advantages
Reply: The problem statement is highlighted in the introduction section (line 90-93). The text was adjusted to include a research hypothesis (line 98-99) and to further highlight the advantages and limitations of our methodology (line 562-569).
Reviewer 3 Report
This paper presents a seven-step workflow for the determination and characterization of vLEs in agricultural landscapes from aerial LiDAR data with evaluation in three study areas in Flanders, Belgium. The presented classification of LiDAR points into the classes "object vLE" and "other non-ground point" was obtained with an overall accuracy between 0.92 and 0.97. Which is a high added value concerning the Logistic Regression model method.
However, in my opinion, the three test objects are not convincing especially that they are very similar to each other. Nevertheless, I find the publication interesting and worthwhile.
I propose:
Add two test areas with different characteristics, more with a varied number of objects on the ground.
Add a magnification to FIG4 E.
Expand the conclusions.
Author Response
We would like to thank the reviewer for their time to assess our manuscript. We have addressed the comments and suggestions made by the reviewer. Below can be found a point-by-point response to the reviewers’ comments and suggestions.
Comment 1: Add two test areas with different characteristics, more with a varied number of objects on the ground.
Reply: Thank you for this suggestion. We believe our three study areas differ sufficiently to demonstrate the transferability of our workflow to other areas. This was already mentioned in the section ‘study area’ (line 103-119) and in Table 1 but the text was adjusted to further highlighted this in line 132-134.
Comment 2: Add a magnification to FIG4 E.
Reply: Figure 4 was adapted to include a magnification of Fig4E.
Comment 3: Expand the conclusions.
Reply: The conclusion section was adapted to also include the outlook for further research along the lines of this paper (line 572-576) and the advantages and limitations of our methodology (line 562-569).